# Imaging the real space structure of the spin fluctuations in an iron-based superconductor

Shun Chi[1,2,*], Ramakrishna Aluru[3,4,*], Stephanie Grothe[1,2,‡], A. Kreisel[5,6], Udai Raj Singh[4], Brian M. Andersen[5], W.N. Hardy[1,2], Ruixing Liang[1,2], D.A. Bonn[1,2], S.A. Burke[1,2,7] & Peter Wahl[3,4]

Spin fluctuations are a leading candidate for the pairing mechanism in high temperature superconductors, supported by the common appearance of a distinct resonance in the spin susceptibility across the cuprates, iron-based superconductors and many heavy fermion materials. The information we have about the spin resonance comes almost exclusively from neutron scattering. Here we demonstrate that by using low-temperature scanning tunnelling microscopy and spectroscopy we can characterize the spin resonance in real space. We show that inelastic tunnelling leads to the characteristic dip-hump feature seen in tunnelling spectra in high temperature superconductors and that this feature arises from excitations of the spin fluctuations. Spatial mapping of this feature near defects allows us to probe non-local properties of the spin susceptibility and to image its real space structure.

[1] Department of Physics and Astronomy, University of British Columbia, Vancouver, British Columbia, Canada V6T 1Z1 . [2] Stewart Blusson Quantum Matter Institute, University of British Columbia, Vancouver, British Columbia, Canada V6T 1Z4. [3] SUPA, School of Physics and Astronomy, University of St Andrews, North Haugh, St Andrews, Fife KY16 9SS, UK. [4] Max-Planck-Institut für Festkörperforschung, Heisenbergstr. 1, D-70569 Stuttgart, Germany. [5] Niels Bohr Institute, University of Copenhagen, Juliane Maries Vej 30, DK-2100 Copenhagen, Denmark. [6] Institut für Theoretische Physik, Universität Leipzig, D-04103 Leipzig, Germany. [7] Department of Chemistry, University of British Columbia, Vancouver British Columbia, Canada V6T 1Z1. * These authors contributed equally to this work. Correspondence and requests for materials should be addressed to P.W. (email: wahl@st-andrews.ac.uk).
‡Deceased.

In phonon-driven strong-coupling superconductors, fingerprints of the pairing boson appear as features outside the gap, arising from a renormalization of the gap function and leading to a characteristic shoulder-dip-hump structure, with the transition between the shoulder and dip occurring at an energy of $\Delta + \Omega$, where $\Delta$ is the superconducting gap and $\Omega$ is the mode energy of the pairing boson (see Fig. 1a, spectrum for lead)[1]. In unconventional superconductors, spin fluctuations might play a similar role in mediating the pairing[2]. In copper oxide[3–8], iron-based[9–11] and many heavy fermion superconductors[12,13], a characteristic resonance is observed by inelastic neutron scattering. The resonance is due to near-nesting with a vector $\mathbf{Q}$ between Fermi surface sheets which exhibit a different sign of the superconducting order parameter (Fig. 1b,c)[5,14,15]. The appearance of strong features in tunnelling spectra of cuprate and iron-based superconductors (see Fig. 1a) at roughly an additional $\Delta$ above the gap, an energy close to the resonance peak in the spin susceptibility (see Fig. 1c, Supplementary Note 1), makes it tempting to assign this to the pairing mode by analogy to conventional superconductors. However, a closer look at these spectra reveals that such an assignment is problematic; evidenced by the distinctly different shape of the above gap features showing a dip-hump character rather than the shoulder-dip-hump seen in conventional superconductors. While the coincidence of the spin resonance energy with above-gap features in tunnelling spectroscopy is alluring, this has not provided the clear fingerprint of the pairing interaction that it did for conventional superconductors. Numerous interpretations have been put forward[16–22] which all involve coupling to an excitation, but are still typically a result of a

renormalization of the electronic structure, impacting the elastic tunnelling channels. Direct excitation of modes, for example vibrational[23] and spin excitations[24,25], can give rise to inelastic tunnelling channels that add spectral weight on top of the elastic tunnelling at and above the mode energy. Early theoretical work invoked inelastic tunnelling to describe the V-shaped gap structure observed in the normal state in cuprate superconductors[26,27]. This idea has recently been applied to the spin resonance in iron-based superconductors[28]. The importance of this contribution for tunnelling spectra obtained in unconventional superconductors, where the excitation spectrum is a continuum rather than a localized mode has yet to be explored experimentally.

Here we show that tunnelling spectra of the clean surface as well as near defects can be consistently described if we account for an inelastic contribution to the tunnelling spectra, exciting the spin resonance mode. We demonstrate spatial mapping of the inelastic contribution as a new path to real space imaging of spin fluctuations.

## Results

**Inelastic contribution to tunnelling spectra.** Figure 2a schematically shows the elastic and inelastic tunnelling processes: at positive bias voltage $V$ applied to the sample,

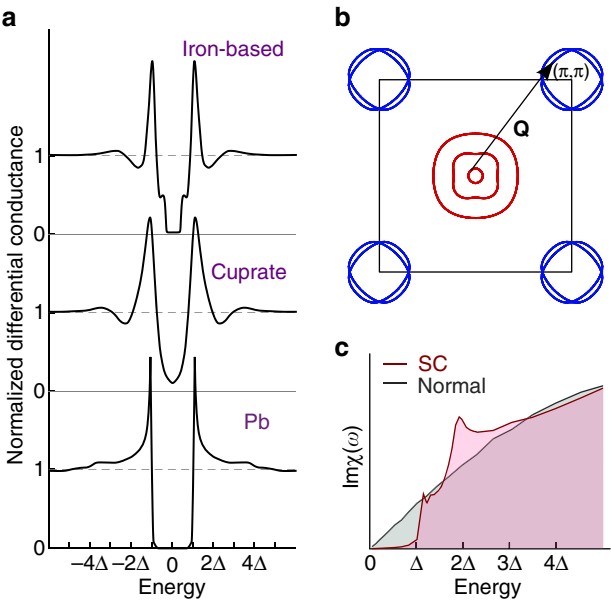

**Figure 1 | Spin excitations in iron-based superconductors. (a)** Schematic tunnelling spectra $g(V) = dI/dV(V)$ obtained on the clean surface of iron-based superconductors and cuprates, showing the dip-hump features. Also shown for comparison is a schematic spectrum for lead (Pb), with the features due to electron-phonon coupling. **(b)** Generic Fermi surface of iron-based superconductors together with the sign of the superconducting order parameter (encoded in colour: $+$ red, $-$ blue). The arrow indicates the nesting vector $\mathbf{Q}$ between hole-like bands at $\Gamma$ and electron-like bands at the zone corner. **(c)** Spin susceptibility for the model with Fermi surface and order parameter shown in **b** (from ref. 32). The spin susceptibility of the normal state electronic structure is rather featureless, whereas it develops a resonance in the superconducting (SC) state which is detected in neutron scattering[34].

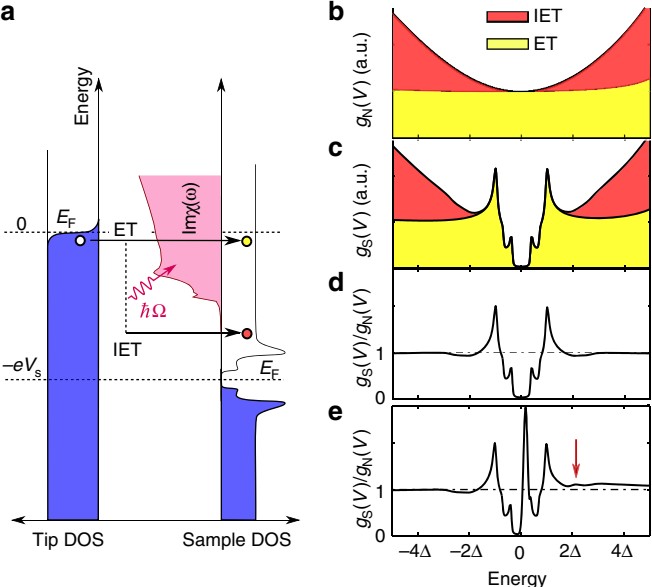

**Figure 2 | Elastic tunnelling (ET) and inelastic tunnelling (IET) in LiFeAs. (a)** Inelastic tunnelling process, which can lead to excitation of the spin resonance at energy $\Omega$ by an electron tunnelling between the tip and the surface (see Supplementary Fig. 2 for a schematic of the inelastic processes involving tunnelling of an electron from the sample to the tip and involving de-excitation). **(b)** Calculated elastic (yellow) and inelastic (red) contributions. The normal-state density of states and spin susceptibility are obtained for a five band tight-binding model (Methods section). **(c)** Calculated elastic and inelastic contributions to the tunnelling current for the superconducting state, using the spin susceptibility for an $s\pm$ order parameter and the known gap structure of LiFeAs. **(d)** Spectrum of the superconducting state normalized by the normal state spectrum. Note the apparent suppression of the differential conductance around $2\Delta$. **(e)** Simulated tunnelling spectrum near a defect which exhibits a sharp defect bound state in the unoccupied states. An additional feature appears due to inelastic tunnelling outside of the superconducting gap at $E_B^\star = E_B + \Omega$, marked by a red arrow. This feature only appears in the unoccupied states, as the defect bound state.

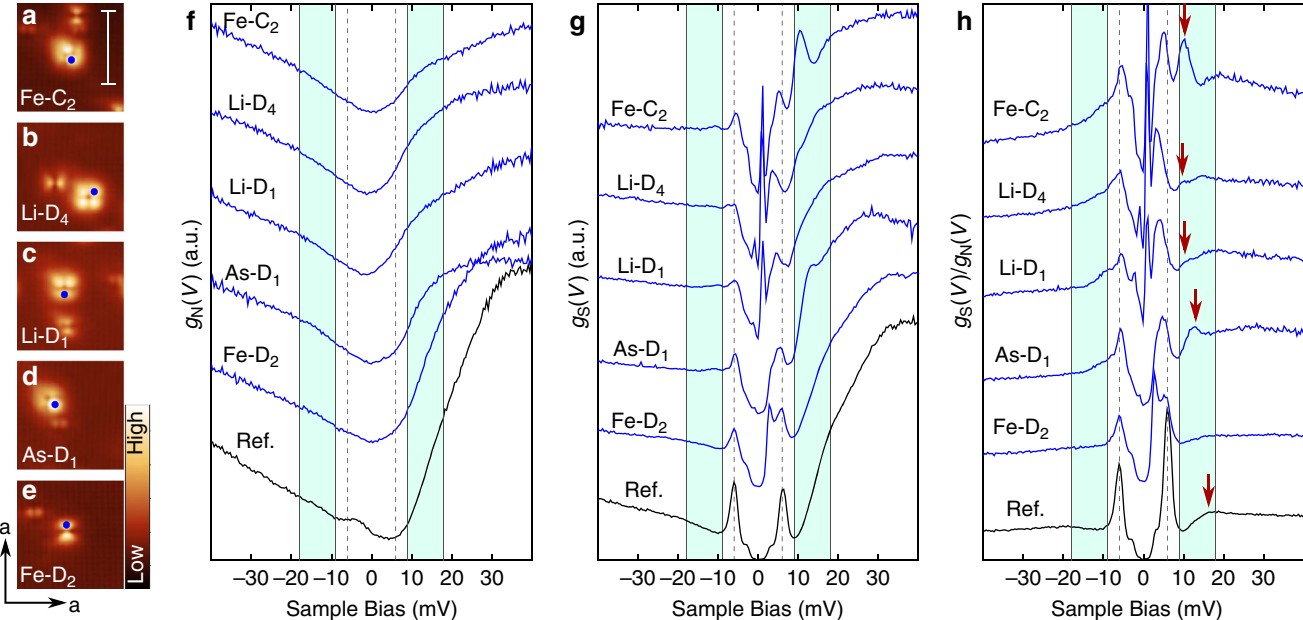

**Figure 3 | Spectra on native defects in LiFeAs.** (**a–e**) topographic images showing the different native defects (measured in the normal state; $V = -50$ mV, $I = 300$ pA, scale bar is 5 nm). The locations where spectra shown in **f–h** were taken are marked by blue dots. Labelling follows the symmetry of the defects as introduced previously[47]; these defects have also been observed by other groups[48]. (**f**) Tunnelling spectra $g_N(V)$ obtained in the normal state ($B = 10$T, $T = 12$ K) and (**g**) $g_S(V)$ in the superconducting state ($T = 1.5$ K) measured on the defects shown to the left. (**h**) Normalized spectra obtained by dividing spectra measured in the superconducting state shown in **g** by the normal state spectra shown in **f**. Spectra and topographies are shown in the same order in **a–h**. Spectra in **f–h** plotted in black (labelled 'Ref') were obtained away from defects on the clean surface. The energy range of the dip-hump feature is marked in cyan in **f–h**. The red arrows in **h** highlight the replica feature in the defect spectra.

an electron tunnelling from the Fermi energy of the tip to the sample can tunnel elastically into a state at energy $eV$ above the Fermi energy of the sample (where $e$ is the elementary charge). This contribution leads to a differential conductance $g_{el}(V) \sim \rho(eV)$ proportional to the density of states (DOS) of the sample $\rho(E)$ (assuming a constant DOS for the tip). Alternatively, if $eV$ is sufficiently large, it can excite a bosonic mode with energy $\Omega$ during the tunnelling process while decaying into a state at energy $eV - \Omega$. The inelastic process is only possible if an unoccupied final state is available (for $V > 0$). The probability for the process is thus proportional to $\mathrm{Im}\chi(\omega)\rho(eV - \omega)f(eV - \omega)$, where $\mathrm{Im}\chi(\omega)$ is the density of bosonic excitations at energy $\omega$ and $f(E)$ the Fermi function. When tunnelling into a metal, the lowest available states are right at the Fermi energy and inelastic features will appear once $|eV| \gtrsim \Omega$. For a single bosonic mode, this leads to a characteristic step in the conductance that is symmetric in bias due to the opening of an additional inelastic tunnelling channel. For a continuous excitation spectrum (Fig. 2b) the total contribution of the inelastic tunnelling to the differential conductance becomes an integral,

$$g_{inel}(V) \propto \int_0^{eV} \mathrm{Im}\chi(\omega)\rho(eV - \omega)\mathrm{d}\omega, \qquad (1)$$

accounting for the Fermi function in the limit $k_B T \ll eV$ ($k_B$: Boltzmann constant, for the full equation including temperature dependence see Supplementary Note 2, Supplementary Equations 8–11 and Supplementary Fig. 2). The total differential conductance is then $g(V) = g_{el}(V) + g_{inel}(V)$. If the imaginary part of the susceptibility, $\mathrm{Im}\chi(\omega)$, exhibits pronounced maxima, replica features will appear in the tunnelling spectra. In inelastic tunnelling into a superconductor, if the density of states and the susceptibility $\mathrm{Im}\chi(\omega)$ are gapped, the inelastic contribution is zero for small voltages $V$,

suppressing the differential conductance relative to the normal state, and becomes largest once the final state lines up with the energy of the coherence peak $eV - \Omega \sim \Delta$. This is shown in Fig. 2c,d for a superconductor with a sign changing order parameter[26–28].

**Defect Spectra.** The dip-hump feature can thus be interpreted as a direct consequence of the inelastic tunnelling process, yielding a continuum of additional channels matching the spin susceptibility. The dip becomes a replica of the superconducting gap, and the hump at $\Delta + \Omega$ is a replica of the coherence peak further enhanced by an increased density in both the excitation spectrum and the final states. Comparison of the simulated tunnelling spectrum (Fig. 2d) with the one obtained on clean LiFeAs (see Fig. 3h, in black spectrum obtained on the clean surface) shows excellent agreement. These spectra obtained on the clean surface are consistent with previous STM experiments[20,29,30], exhibiting two superconducting gaps at $\Delta = 6$ meV and $\Delta_{small} = 3$ meV as well as the pronounced dip-hump structure.

In a similar way, sharp defect bound states at energy $E_B$ will lead to replica features at $E_B^\star = E_B + \Omega$ (Fig. 2e). To test this expectation, we have analysed spectra obtained near five different types of native defects in as-cleaved LiFeAs (see Fig. 3a–e, Table 1). While the normal state spectra $g_N(V)$ (Fig. 3f) of these defects are rather featureless, the spectra $g_S(V)$ obtained in the superconducting state show pronounced in-gap bound states (Fig. 3g). In addition to the in-gap bound state, the spectra show clear maxima at energies larger than the gap size (Fig. 3h). The in-gap bound states are well reproduced by $T$-matrix calculations using a five band model for the band structure and spin-fluctuation mediated pairing[31,32]. The calculations, which do not account for the inelastic tunnelling channel, do not show any characteristic resonance feature at an energy scale beyond the superconducting gap, irrespective of whether a purely potential or

**Table 1 | Summary of defect-bound states and replica features observed on native defects in LiFeAs.**

|  | $E_B$ (meV) | $E_B^{\star}$ (meV) | $\Omega$ (meV) |
|---|---|---|---|
| Fe-$C_2$ | 0.86 | 10.3 | 9.4 |
| Li-$D_4$ | 0.86 | 10.3 | 9.4 |
| Li-$D_1$ | 1.1 | 10.9 | 9.8 |
| As-$D_1$ | 4.86 | 14 | 9.1 |
| Fe-$D_2$ | 2.76 |  |  |
| clean | 6 | 15.9 | 9.9 |
| Average |  |  | $9.5 \pm 0.3$ |

$E_B$ refers to the bound state at positive bias voltage and $E_B^{\star}$ to the energy of the replica feature. For the clean surface, we quote the energy $\Delta$ of the coherence peak of the larger gap as $E_B$ and its replica feature as $E_B^{\star}$.

magnetic scatterer is considered. Yet, the fact that the out-of-gap feature disappears when the material undergoes the transition into the normal state unequivocally links it to superconductivity. Calculations of tunnelling spectra within the model sketched in Fig. 2, accounting for the inelastic contribution and including the in-gap bound state, reproduce the measured spectra well (Fig. 2e). In particular, the replica features due to inelastic tunnelling are only expected to occur on the same side of the Fermi energy (zero bias) as the defect bound state itself, fully consistent with the experimental observation of both only at positive bias voltages. Analysing the energy $\Omega = E_B^{\star} - E_B$ of the spin resonance mode from the bound state replica features observed on the defects, we obtain $\Omega = 9.5 \pm 0.3$ meV, with an astonishingly small scatter of the values and in very good agreement with the enhanced weight of spin fluctuations observed in the superconducting state by inelastic neutron scattering at energy transfer of 8 meV (refs 33–35). We therefore identify these features as the replica features of the impurity bound states due to inelastic tunnelling.

**Spatial Evolution of replica feature.** From differential conductance maps, we have analysed the spatial evolution of the bound state as well as that of the replica features, see Fig. 4. While the in-gap bound state (marked by a black arrow) is localized within 1 nm of the defect site, the replica feature (marked by a red arrow in Fig. 4), persists over substantially larger distances from the defect, and can be observed up to almost 3 nm from the defect site. This is more clearly seen in real space maps of the bound state and the replica feature, see Fig. 5a,b. A quantitative analysis of the spatial decay (Fig. 5c) for the Fe-$C_2$ defect reveals that the length scale over which the replica feature is observed is much larger than for the in-gap bound state and that the nature of the decay differs. The in-gap bound state decays exponentially with a characteristic decay length, whereas the spatial dependence of the replica feature is well described by a $1/r$ behaviour, as expected for the geometrical factor for point-like scattering of a two-dimensional state. Similar to Equation 1, which describes local inelastic excitations, a non-local contribution is expected to be of the form

$$g_{inel}(\mathbf{r}, V) \propto \int_0^{eV} \int \rho(\mathbf{r}', eV - \omega) \mathrm{Im}\chi(\mathbf{r}, \mathbf{r}', \omega) d\mathbf{r}' d\omega. \quad (2)$$

for an impurity at the origin ($\mathbf{r} = 0$), the real space susceptibility $\mathrm{Im}\chi(\mathbf{r}, \mathbf{r}', \omega)$ and bound state density of states $\rho(\mathbf{r}', \omega)$. This process is schematically depicted in Fig. 5d. The energy of the replica feature within the dip-hump structure is constant apart from a small spatial dependence which can be attributed to variations in the normal state DOS (Supplementary Note 3, Supplementary Figs 3 and 4).

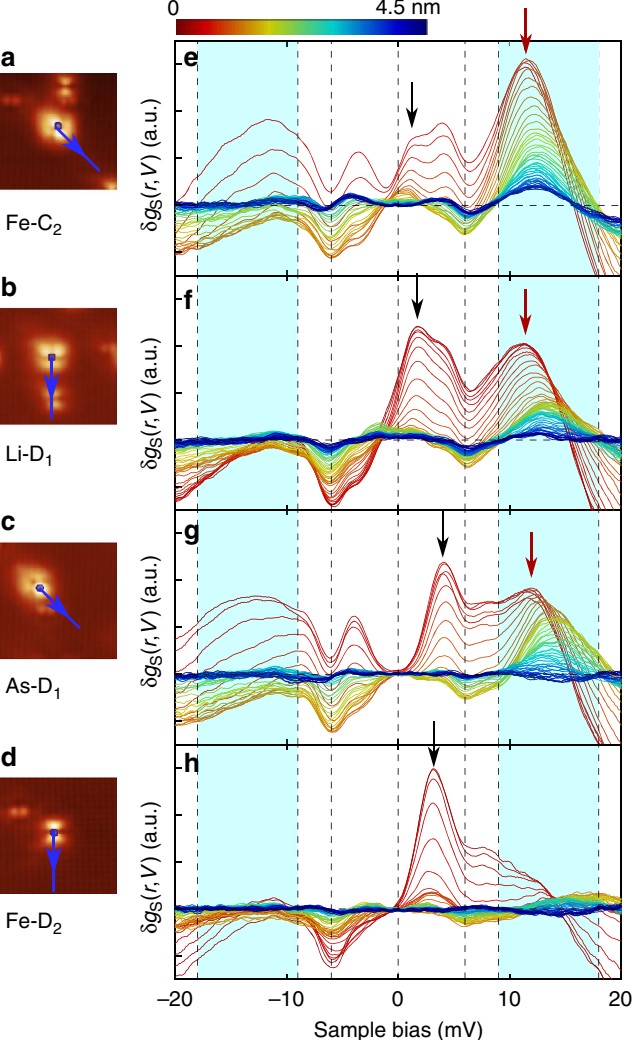

**Figure 4 | Spatial evolution of the tunnelling spectra.** (**a**–**d**) topographies and (**e**–**h**) difference spectra $\delta g_S(r,V) = g_S(r,V) - g_S(r \rightarrow \infty, V)$, where $r$ is the distance to the defect. The difference spectra ($T = 4.2$ K) are shown from on top of the defects (red) to 4.5 nm away from the defect (dark blue). The arrows in the topographies mark the directions of the line cuts. The defect bound states are marked by black arrows. The peaks in the dip-hump energy window (cyan area) are marked by red arrows. The peaks in the dip-hump energy window are visible to significantly larger distance from the defect than the defect bound states.

## Discussion

An interpretation based on inelastic tunnelling alone, neglecting renormalization of the electronic structure due to electron-boson interactions, already yields excellent agreement when considering a spin susceptibility in a sign-changing superconducting state[28,36]; that is, $s\pm$. The energy of the dip-hump structure is consistent with that of the spin resonance detected in neutron scattering[33,34]. A non-sign changing order parameter does not yield a resonance in the spin susceptibility and hence the spectra would not exhibit a strong dip-hump feature (compare Supplementary Fig. 5). The features produced by inelastic tunnelling provide detailed insight into the spectrum of the underlying spin excitations, which in turn provide tell-tale signatures of the superconducting order parameter (see also Supplementary Note 4, Supplementary Figs 5 and 6). The observation of the replica feature beyond the length scale of the in-gap bound state shows that inelastic tunnelling can probe non-local properties of the spin resonance.

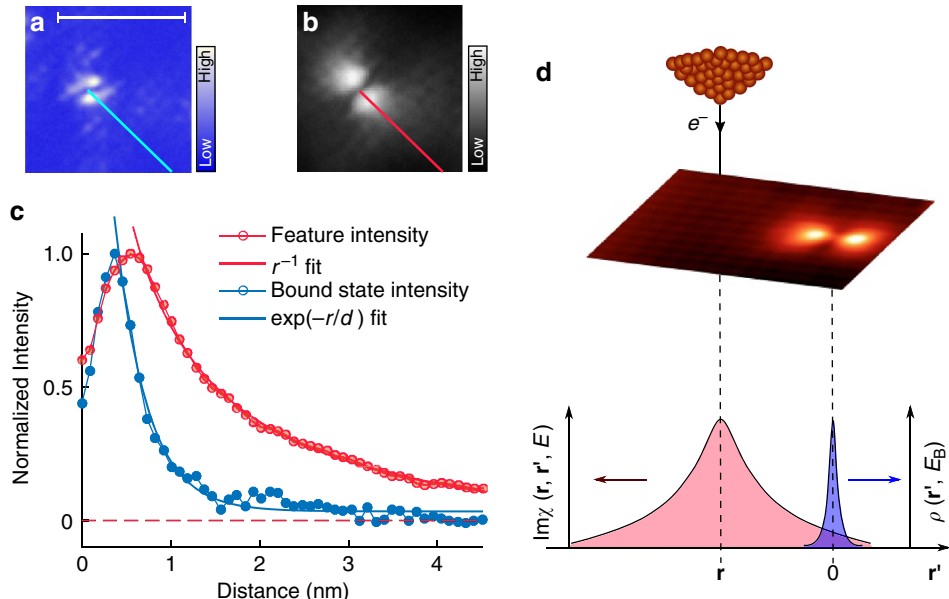

**Figure 5 | Imaging of spin excitations. (a)** Map of the conductance in the superconducting state $g_S(\mathbf{r},V)$ at the energy of the defect bound state for an Fe-C$_2$ defect (scale bar, 5 nm). **(b)** Map of the peak height of the replica feature for the same defect, obtained by tracking the peak intensity in the energy range of the replica feature - providing a real-space image of the spin excitations. **(c)** Spatial evolution of the defect bound state (blue) and the replica feature within the dip-hump energy range (red) from the center of the defect to the clean surface. The values are normalized to the maximum value for each of the two curves. The cut along which the distance dependence has been extracted is shown as a blue and red solid line in **a,b**, respectively. **(d)** Sketch of the non-local inelastic excitation process: an electron from the tip interacts with the spin fluctuations, which decay at the position of the defect. The spatial structure of the spin fluctuations and the defect state are shown as red and blue curves.

Non-local effects in inelastic tunnelling have been discussed previously in the context of molecules interacting with the surface state of noble metal surfaces[37]. The present experimental conditions differ from this, since the inelastic mode is not localized, but has a spatially extended structure, and only the final state is localized at the defect. Within the non-local mechanism proposed for vibrational excitations[37], the spatial structure of the replica feature in conductance maps would produce signatures similar to quasi-particle scattering and contain information about its wave vector - which we do not observe (Fig. 5c). Because the spin excitations are based on an electronic degree of freedom, the excitation by the tunnelling electron and its release into the final state may occur at spatially separated locations. This provides a new mechanism for non-local inelastic tunnelling and thus reveals information about the real-space structure of the spin resonance mode. The inelastic tunnelling process will also have implications for the interpretation of quasi-particle interference imaging[38], because it introduces additional contributions to the differential conductance. The decay length then is a property of the spin susceptibility. From the width (FWHM) of the spin resonance observed in neutron scattering at wave vector $Q_0$, $\delta_q \sim 0.1 Q_0$, we can extract an estimate for the spatial correlation length of the spin fluctuations on the order of $\frac{2\pi}{\delta_q} \sim 3.8$ nm, which compares quite well with the spatial extent reported here. Also, comparison with the calculated real space structure of the spin susceptibility (compare Supplementary Fig. 1a) yields good agreement of the characteristic length scale.

Our results have relevance beyond sign-changing super-conductors. They raise more broadly the question of what the impact of spin fluctuations is on the tunnelling spectra of strongly correlated electron materials, and whether indeed pseudo-gap shaped spectra might have contributions from inelastic spin excitations. In topological insulators and materials with topologically non-trivial spin textures across the Fermi surface, an enhanced spin susceptibility is also observed[39] and could facilitate

imaging of magnetic fluctuations near defects[40]. The combination of inelastic tunnelling to characterize spin excitations with the atomic resolution capability of spin-polarized STM to resolve the magnetic structure of quantum materials[41] will provide new insights into the impact of defects on emergent orders[42,43]. Our measurements thus provide access to a real space picture of resonant spin excitations in unconventional superconductors, which we demonstrate specifically for LiFeAs. Inelastic tunnelling is shown to fully account for the dip-hump features in this material. It is expected to play a similarly important role in other unconventional superconductors, for example in the cuprates, where the interpretation of the dip-hump features in tunnelling spectra has been highly controversial[16,44]. In all sign-changing superconductors characteristic dip-hump features are expected to be present in tunnelling spectra at energies larger than the superconducting gap magnitude. Although the dip-hump feature cannot be taken as *prima facie* evidence of spin-mediated pairing, it is evidence of a sign-changing order parameter.

## Methods

**Sample Growth.** LiFeAs single crystals were grown using a self-flux method[20]. Data was taken on three different samples[32] - LiFeAs, LiFe$_{0.998}$Mn$_{0.002}$As, and LiFe$_{0.997}$Ni$_{0.003}$As. The change of tunnelling spectra caused by the minimal substitution levels are negligible[32]. The measured native defects are well separated from each other and from the substituted elements.

**STM Experiments.** Two different scanning tunnelling microscopes were used throughout this study. A home-built low temperature scanning tunnelling microscope (STM) operating at temperatures down to 1.5 K (ref. 45), as well as a commercial STM manufactured by Createc with a base temperature of 4.2 K. The normal state spectra in Fig. 3f were measured in high magnetic field to reduce the thermal broadening effect, using the fact that $T_c$ of LiFeAs is suppressed from 17 K ($k_B T \sim 1.5$ meV) to 12 K ($k_B T \sim 1.0$ meV) in a 10 T magnetic field. In defect-free regions, there is no evident difference between the $g(V)$ spectra acquired at 17 K (ref. 20) and the spectra obtained at 12 K and in 10 T magnetic field shown here. Therefore, the perturbation of the 10 T magnetic field on the normal state DOS is negligible.

**Theory.** Simulations of inelastic tunnelling spectra following Equation (1) have been performed in the normal state and the superconducting state, where the bosonic spectrum was obtained from a calculation of the local susceptibility $Im\chi(\omega) = \sum_{\mathbf{q}} Im\chi_{RPA}(\omega, \mathbf{q})$. For the full equation for the differential conductance and more details on the calculations see Supplementary Notes 1 and 2. For the normal state, the susceptibility was calculated as the generalization of the Lindhard function to a five band model[32,46] together with the inclusion of interactions from a Hubbard-Hund Hamiltonian in the random phase approximation (RPA), yielding a featureless bosonic spectrum as shown in Fig. 1c. In the superconducting state, the superconducting order parameter in reciprocal space as sketched in Fig. 1b was included via coherence factors giving rise to a strong resonance at approximately $\Omega \approx 2\Delta$ and a weak resonance at $\Omega \approx \Delta + \Delta_{small}$.

**Data availability.** Underpinning data will be made available at http://dx.doi.org/10.17630/43ff2e1e-36d4-4c4b-b987-681ec3de3fe8.

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

## Acknowledgements

We acknowledge useful discussions with Steve Johnston, Peter Hirschfeld and Morten H. Christensen. This project has been partially funded by the MPG-UBC center. BMA and AK acknowledge funding from Lundbeckfond (Grant No. A9318) and PW by EPSRC under EP/I031014/1. Work at UBC was supported by the Natural Sciences and Engineering Research Council of Canada (grants 402072-2012, 170825-13), the Canada Research Chairs Program, the Canadian Institute for Advanced Research, and the Stewart Blusson Quantum Matter Institute.

## Author contributions

S.C., R.A., S.G. and U.R.S. carried out STM experiments; S.C. and R.A. analysed the data; S.C., W.N.H., R.L. and D.A.B. grew the crystals; A.K. and B.M.A. performed the calculations of the spin excitations, P.W. and S.C. wrote the manuscript with help of S.A.B., R.A. and D.A.B. All authors discussed and contributed to the manuscript.
