## [Peer review file · Nature Communications]

Reviewers' comments:

Reviewer #1 (Remarks to the Author):

The authors investigate the signatures of the magnetic resonance modes in iron-based superconductors in the differential conductance measured near defects in LiFeAs. They argue that the resonance mode should lead to a replica effect, such that an impurity state at energy E_b would immediately lead to an additional peak in dI/dV at energy $E_b + \Omega$, where Ω is the energy of the resonance mode.

Overall, I like the idea that one can use the STM spectra near defect to identify signature of the collective modes, and in particular the neutron resonance through the generation of replica effects. However, I think that there are a number of shortcomings to this article, such that I would like the authors to address the points below before making a final recommendation regarding the publication of this manuscript.

1) The authors state on p.4:

"Comparison of the simulated tunneling spectrum (Fig. 2d) with those obtained on LiFeAs (see Fig. 3c) show excellent agreement. "

I could not find any discussion in the article or the SI of how this simulation was performed, and the authors should address this. The authors also refer to these calculations on p. 5, where they state

"Calculations of tunneling spectra within the model sketched in Fig. 2, accounting for the inelastic contribution and including 88 the in-gap bound state, reproduce the measured spectra well (Fig. 2e). "

without giving any details, neither in the main text nor in the SI. Also, did the authors mean Fig. 2e, rather than Fig. 2d, as Fig. 2d does not contain any defect state? Finally, I am not sure that I would agree with the authors' assertion that there is excellent agreement between the theory and the experimental data in Fig. 3c. The latter are shown over a rather wide energy range, such that it is rather difficult to see whether the difference between E_b and the replica peak is indeed a constant for all cuts.

2) In general, impurities induce defect states that have both a particle-like and a hole-like branch, meaning that there should be peaks inside the SC gap at $+E_b$ and at $-E_b$. However, the authors seem to exclusively focus on the hole-like branch. Is there no negative energy, particle-like branch of the defect state? And if there is, is there also a replica of that state visible on the negative energy side of the spectrum?

3) I could not find any analysis of the boson excitation energy in the article. What is the value of this energy, and is it consistent with the one measured for the neutron resonance? Is this energy constant (as one would expect) in the spectra for different defects?

4) The authors state that on p. 6

"The features produced by inelastic tunneling provide detailed insight into

the spectrum of the underlying spin excitations, which in turn provide tell-tale signatures of the superconducting order parameter. "

This is an interesting idea, but what is the information on the spectrum of the spin excitations that the authors extracted from their data? As mentioned above in 3), I would expect that the simplest quantity to extract would be the bosonic energy, but I could not find it in the paper. As this piece of information is also known from neutron scattering data, what other, new information can be extracted from the STM data?

5) After Eq.(S3) in the SI, the authors talk about simulated spectra in Fig.2b. I do not understand how these simulated data were generated. The authors should explain this. Are these simulations based on computing dI/dV from some starting Hamiltonian? If so, this needs to be explained in detail.

6) In Eq.(S4), the left side is the imaginary part of the real space susceptibility.

Reviewer #2 (Remarks to the Author):

The manuscript of Chi et al. addresses the pairing mechanism in LiFeAs. In the superconducting state, fundamental excitations are detected in close proximity of defects. These are interpreted as a signature of spin-fluctuations taking place in the compound.

Contrary to many previous studies, the technique used by the author (STM/STS) has a fundamental advantage: it allows to visualize the real space structure of fundamental phenomena as a function of defects and, more generally, inhomogeneities. I believe this work has the potential for having high impact, especially if this approach can be extended and applied to other systems which are also characterized by impurity states in the gap (see comment below).

For this reason, i.e. directly tackling a scientifically relevant open question using an original approach which sheds light on new intriguing aspects, I think the manuscript deserves publication in Nature Communications.

The manuscript is nicely written and it was a pleasure to read. However, before publication, there is a major aspect which I think the authors should carefully address.

Major:

The inelastic excitations seen in the spectra strongly depend on the presence of defects in the nearby. It is thus very important to understand the origin of the defects. In the present state, the authors assign every feature visible in Fig. 3 (left column) to well defined defects. However, I find this assignment pretty ambiguous. This is far from being state of the art. Theoretically calculated STM images are nowadays routinely performed for defects assignment. If this is not possible, the authors should at least carefully refer to existing works (see for example Phys. Status Solidi B 254, No. 1, 1600159 (2017)).

Minor:

1) I would suggest the authors to modify the introduction with the goal of putting their work in a

more broad perspective. Impurity states play a major role in many condensed matter systems of timely interest. Well beyond the cuprates cited by the authors themselves. See for example graphene Chemical Physics Letters 476 (2009) 125 or topological insulators NatComm7, 12027 (2016).

If the technique used by the authors can be extended to these systems it would certainly be a major advance in the field. For these reasons, I recommend the authors to put this "impurity assisted visualization of fundamental excitations" in a more broad context.

2) The authors show that inelastic excitations decay slower than the bound state. This should then make possible to visualize the emergence of possible inelastic interference phenomena as suggested in PRB 85, 161401(R). The author might consider this aspect for future experiments.

Reviewer #3 (Remarks to the Author):

The manuscript entitled "Imaging the Real Space Structure of the Spin Fluctuations in an Iron-based superconductor" by Shun Chi et al. reports scanning tunneling microscope measurements on LiFeAs containing various impurities. By comparing the spectra obtained at the impurities with those recorded at clean positions on the surface the authors

claim to obtain information about the spatial extension of magnetic fluctuations, that otherwise is very difficult to get. This work extends their own recent Physical Review B article on the same compounds, in which the appearance of bound states at impurities was observed and interpreted by a thorough theoretical analysis to indicate a s_{\pm} character of the superconducting order. Here, emphasis is put on - mostly weak - replica features appearing at a constant energy above these bound states. The authors attribute the replica features to the spin resonance mode, and from the spatial dependence of this signal they claim to obtain the spatial extension of the resonance excitations.

The identification of this signal as the spin-resonance mode seems, however, insufficiently supported by the data and the theoretical analysis. For LiFeAs a spin resonance at such energy has not been found in the various neutron scattering experiments. There is little signature in the magnetic response at this energy, which contrasts with the sharp signal reported in this STM experiment (in particular the upper spectrum of Fig. 2b). Can the authors exclude other explanations for the sometimes sharp signal in the spectra of Fig. 2b). There might exist a magnetic mode that is directly coupled to the impurity. Is it possible that the Fe-C2 spectra sense something different from the others, as only here a real peak can be separated, while the other spectra only exhibit a shoulder. A better analysis of the spectra by fitting positions and strengths of the various signals might help future analyzes. If the authors want to keep their conclusion that this replica feature stems unambiguously from a magnetic excitations, additional arguments should be given.

Since I am not convinced about the unambiguous identification of the origin of the replica features as magnetic modes, the discussion about the spatial extension seems speculative.

When modeling the spectra, the proportionality factor entering the inelastic scattering compared to the elastic one seems crucial and should be discussed.

In summary I may not recommend the manuscript in its present form for publication in nature communications due to its somehow speculative conclusion. I also wonder whether the mapping of the spin resonance excitation pinned to an impurity can easily be transferred to that in the bulk material.

Reviewers' comments:

Reviewer #1 (Remarks to the Author):

The authors investigate the signatures of the magnetic resonance modes in iron-based superconductors in the differential conductance measured near defects in LiFeAs. They argue that the resonance mode should lead to a replica effect, such that an impurity state at energy E_b would immediately lead to an additional peak in dI/dV at energy $E_b + \Omega$, where Ω is the energy of the resonance mode.

Overall, I like the idea that one can use the STM spectra near defect to identify signature of the collective modes, and in particular the neutron resonance through the generation of replica effects. However, I think that there are a number of shortcomings to this articles, such that I would like the authors to address the points below before making a final recommendation regarding the publication of this manuscript.

1) The authors state on p.4:

"Comparison of the simulated tunneling spectrum (Fig. 2d) with those obtained on LiFeAs (see Fig. 3c) show excellent agreement. "

I could not find any discussion in the article or the SI of how this simulation was performed, and the authors should address this. The authors also refer to this calculations on p. 5, where they state

"Calculations of tunneling spectra within the model sketched in Fig. 2, accounting for the inelastic contribution and including 88 the in-gap bound state, reproduce the measured spectra well (Fig. 2e). "

We have added supplementary section S2 and supplementary fig. S2 and added details to supplementary section S1 (formerly supplementary section S3) which provide more details on the theory, i.e. the calculation of the spin resonance and the calculation of the differential conductance. The calculation of the spin susceptibility builds on previous work (refs. 28, S4), also the equations for the inelastic contribution to the tunneling current have been derived previously (refs. 24-26).

without giving any details, neither in the main text nor in the SI. Also, did the authors mean Fig. 2e, rather than Fig.2d, as Fig.2d does not contain any defect state? Finally, I am not sure that I would agree with the authors assertion that there is excellent agreement

between the theory and the exp. data in Fig. 3c. The latter are shown over a rather wide energy range, such that it is rather difficult to see whether the difference between E_b and the replica peak is indeed a constant for all cuts.

To highlight this point, we have added in table 1 the energy of Omega extracted for the defect bound states and the replica features. As can be seen, they consistently all fall into an energy range of 9-10 meV. The spin-fluctuation resonance observed by inelastic neutron scattering measurements (refs. 31-33) is in the range of 6-10 meV, in agreement with our result. We have added a discussion in lines 85-89.

2) In general, impurities induce defect states that have both a particle-like and a hole-like branch, meaning that there should be peaks inside the SC gap at $+E_b$ and at $-E_b$. However, the authors seem to exclusively focus on the hole-like branch. Is there no negative energy, particle-like branch of the defect state? And if there is, is there also a replica of that state visible on the negative energy side of the spectrum?

Both experimentally, and theoretically, the spectral weight of the defect bound states at negative energies is found to be small [refs. 17, 35, 47].

We don't have an explicit explanation why this is happening, but it is clear that it is an effect of the normal state electronic structure not being particle-hole symmetric (as can be seen from tunneling spectra obtained in the normal state, see fig. 3a) together with the impurity being a potential scatterer with a certain sign of the impurity potential. Thus, this is due to details of the electronic structure.

We note that for two of the impurities (Fe-C2, Li-D1), one can also see a bound state and a shoulder at negative energies indicating that there is also a replica feature if there is a sufficiently strong and sharp bound state closer to the Fermi energy in the occupied states.

3) I could not find any analysis of the boson excitation energy in the article. What is the value of this energy, and is it consistent with the one measured for the neutron resonance? Is this energy constant (as one would expect) in the spectra for different defects?

Following the suggestion of the referee, we have added to table 1 the energy of the replica feature of the clean surface, as well as the energy of the excitation for each of the systems investigated here and the average excitation energy. We have added a discussion of the comparison to the neutron resonance (lines 85-89). The excitation energy found here is highly consistent across the different defects.

4) The authors state that on p. 6

"The features produced by inelastic tunneling provide detailed insight into the spectrum of the underlying spin excitations, which in turn provide tell-tale signatures of the superconducting order parameter. "

This is an interesting idea, but what is the information on the spectrum of the spin excitations that the authors extracted from their data? As mentioned above in 3), I would expect that the simplest quantity to extract would be the bosonic energy, but I could not find it in the paper. As this piece of information is also known from neutron scattering data, what

other, new information can be extract from the STM data?

We are grateful for the referee to point this out. While the spin fluctuation spectrum is continuous, to replicate the experimental results it needs to have a sharp, resonant feature. We have added the energy of this resonant feature extracted from our data (line 86, 87) in the main text and in table I. Beyond the bosonic energy, we are able to map out the spatial variation of the bosonic excitation close to defects, complementary to the momentum space information obtained by inelastic neutron scattering. This yields consistent energies and decay lengths (as discussed in lines 85-89, 149-153) between the two methods. More importantly, we present a new way to measure bosonic excitations in quantum materials (not limited to spin fluctuations) by localized defect bound states and often with new methods come new insights. This is relevant to many strongly correlated electron materials where spin fluctuations are important, e.g. near a quantum critical point, but might also be relevant for topological insulators or graphene. Furthermore, our results highlight a new way to extract information (albeit indirect) about the phase of the superconducting condensate from tunneling spectra.

5) After Eq.(S3) in the SI, the authors talk about simulated spectra in Fig.2b. I do not understand how these simulated data were generated. The authors should explain this. Are these simulations based on computing dI/dV from some starting Hamiltonian? If so, this needs to be explained in detail.

We have added details of the calculation of the spin susceptibility and the inelastic contribution to the differential conductance to the supplementary material, supplementary methods 1 and 2, eqs. S1-6, 8-11 and fig. S2 (see also above).

6) In Eq.(S4), the left side is the imaginary part of the reals space susceptibility.

We thank the referee for pointing this out. We have followed the convention of Hlobil et al. (arXiv:1603.05288) and defined $\chi(r,\omega)$ as the imaginary part only, we have added this in the text (see supplementary material) and updated the figures accordingly (fig. 1, S2).

Reviewer #2 (Remarks to the Author):

The manuscript of Chi et al. addresses the pairing mechanism in LiFeAs. In the superconducting state, fundamental excitations are detected in close proximity of defects. These are interpreted as a signature of spin-fluctuations taking plane in the compound.

Contrary to many previous studies, the technique used by the author (STM/STS) has a fundamental advantage: it allows to visualize the real space structure of fundamental phenomena as a function of defects and, more generally, inhomogeneities. I believe this work has the potential for having high impact, especially if this approach can be extended and applied to other systems which are also characterized by impurity states in the gap (see comment below).

For this reason, i.e. directly tackling a scientifically relevant open question using an original approach which sheds light on new intriguing aspects, I think the manuscript deserves publication in NatureCommunications.

The manuscript is nicely written and it was a pleasure to read. However, before publication, there is a major aspect which I think the authors should carefully address.

Major:

The inelastic excitations seen in the spectra strongly depend on the presence of defects in the nearby. It is thus very important to understand the origin of the defects. In the present state, the authors assign every feature visible in Fig. 3 (left column) to well defined defects. However, I find this assignment pretty ambiguous. This is far from being state of the art. Theoretically calculated STM images are nowadays routinely performed for defects assignment.

If this is not possible, the authors should at least carefully refer to existing works (see for example Phys. Status Solidi B 254, No. 1, 1600159 (2017)).

The precise nature of these defects is not known. While for known defects replacing Fe sites, a first principles based calculation of topographies and maps as measured in STM is possible, attempts to do the same for the native defects (that are mainly discussed in the present work) are more difficult. Reasons are the large parameter space associated with the different defect types and the requirement of local relaxation close to the defect sites. The defects and the defect spectra are however highly reproducible and have been seen by a number of groups (Hanaguri (unpublished), refs 35, 43, 44). For the purposes of the argument put forward here, the precise nature of the defects is not relevant, merely the fact that they show in-gap bound states plus out-of-gap features which are not captured by theory of defects in superconductors (see refs. 34, 35). We have added the reference suggested by the referee (ref. 44 of the revised manuscript, caption fig. 3). We have further added a sentence to highlight the consistency of experimental results with previously published work (lines 79-82 and caption fig. 3, refs. 17, 29, 30, 35, 46, 47).

Minor:

1) I would suggest the authors to modify the introduction with the goal of putting their work in a more broad perspective. Impurity states play a major role in many condensed matter systems of timely interest. Well beyond the cuprates cited by the authors themselves. See for example graphene Chemical Physics Letters 476 (2009) 125 or topological insulators NatComm7, 12027 (2016).

If the technique used by the authors can be extended to these systems it would certainly be a major advance in the field. For these reasons, I recommend the authors to put this "impurity assisted visualization of fundamental excitations" in a more broad context.

We thank the referee for this suggestion. We have added a paragraph discussing this point in the manuscript (lines 156-162), where we also cite the work mentioned by the referee (NatComm7, 12027, ref. 40). We also have included a sentence of

the opportunities of using inelastic tunneling spectroscopy for quantum materials (lines 162-164, refs. 41-43).

2) The authors show that inelastic excitations decay slower than the bound state. This should then make possible to visualize the emergence of possible inelastic interference phenomena as suggested in PRB 85, 161401(R). The author might consider this aspect for future experiments.

We thank the referee for this suggestion. We had, in fact, a discussion of this effect in an earlier version of the manuscript which we had removed due to constraints on the length, but have introduced it again in the revised version (lines 137-143, ref. 37). We also added a discussion of the potential relevance for quasi-particle interference imaging (lines 143-149, ref. 38). We believe, however, that the dominant effect we observe for the larger decay length of the replica feature is not due to quasi-particles propagating to the defect and decaying there, as discussed in the revised version, because we do not observe a characteristic wave vector of the decay.

Reviewer #3 (Remarks to the Author):

The manuscript entitled "Imaging the Real Space Structure of the Spin Fluctuations in an Iron-based superconductor" by Shun Chi et al. reports scanning tunneling microscope measurements on LiFeAs containing various impurities. By comparing the spectra obtained at the impurities with those recorded at clean positions on the surface the authors claim to obtain information about the spatial extension of magnetic fluctuations, that otherwise is very difficult to get. This work extends their own recent Physical Review B article on the same compounds, in which the appearance of bound states at impurities was observed and interpreted by a thorough theoretical analysis to indicate a s_{\pm} character of the superconducting order. Here, emphasis is put on - mostly weak - replica features appearing at a constant energy above these bound states. The authors attribute the replica features to the spin resonance mode, and from the spatial dependence of this signal they claim to obtain the spatial extension of the resonance excitations.

The identification of this signal as the spin-resonance mode seems, however, insufficiently supported by the data and the theoretical analysis. For LiFeAs a spin resonance at such energy has not been found in the various neutron scattering experiments.

We thank the referee for pointing this out, and have clarified this in the revised manuscript. Inelastic neutron scattering (INS) shows a resonance peak at $\sim 6-10$ meV (refs. 31-33), consistent with what we observe (~ 10 meV difference between the replica and the bound states for all defects). To highlight this point, we have added the energy of the spin excitation in table I, as well as a sentence discussing explicitly the comparison to neutron scattering (lines 85-89). In addition, ref. 33 has examined and confirmed that the INS signal has a magnetic origin. Most other iron

pnictide superconductors exhibit similar spin fluctuations, often stronger than in LiFeAs, where a resonance appears below T_c and within an energy range between Δ and 2Δ (refs. 8-10, 28).

There is little signature in the magnetic response at this energy, which contrasts with the sharp signal reported in this STM experiment (in particular the upper spectrum of Fig. 2b). Can the authors exclude other explanations for the sometimes sharp signal in the spectra of Fig. 2b). There might exist a magnetic mode that is directly coupled to the impurity. Is it possible that the Fe-C2 spectra sense something different from the others, as only here a real peak can be separated, while the other spectra only exhibit a shoulder.

The strength of the replica of the bound states will depend on a combination of the tunneling matrix element between tip and the electronic states near the Fermi level, the coupling between the bound state excitation and the spin fluctuations and the precise spectral form of the bound state as well as of the spin fluctuation spectrum. With regards to the tunneling matrix element, while we don't observe a strong tip dependence, bands with, e.g., d_{xy} or $d_{x^2-y^2}$ character couple in general less to the tip compared to bands with d_{xz} , d_{yz} or d_{z^2} character. Because the bands near the Fermi energy in LiFeAs are predominantly of d -character, depending on the dominant orbital character of the defect bound state, certain bands will couple less to the bound state than others leading to different strengths of the replica feature.

While the intensities differ strongly between different types of defect, we show in Table 1 that the energy differences between the replica and the bound states are very consistent ~ 10 meV. In addition, there are no peak/shoulder features in spectra obtained in the normal state, above T_c , indicating the peak/shoulders are associated with superconductivity. So far, there is no experimental evidence or theoretical proof that either a magnetic or non-magnetic defect would induce peaks/shoulders above but near the superconducting gaps (see, e.g., refs. 34, 35 of the manuscript). Inelastic tunneling naturally explains the consistent energy differences for all defects and agrees with the inelastic neutron scattering results (refs 31-33).

A better analysis of the spectra by fitting positions and strengths of the various signals might help future analyzes. If the authors want to keep their conclusion that this replica feature stems unambiguously from a magnetic excitation, additional arguments should be given.

As pointed out above, we have added a more detailed analysis in table I and the text. We don't think that fitting the replica positions will add anything here, as the strength will depend on matrix elements which are yet poorly understood and the positions can readily be extracted from the spectra. We would like to point out that fig. 5b does contain a detailed map of the intensity of the replica feature as a

function of position, extracted from a whole map.

To cross-check that our analysis yields robust results, we have performed a fit of the replica feature using a Gaussian for the defect bound state of an Fe-C2 defect (see graph below), which yields an energy of the replica feature of 10.2meV, compared to 10.3meV when using the peak energy as in the analysis of the other defects in the main manuscript.

Since I am not convinced about the unambiguous identification of the origin of the replica features as magnetic modes, the discussion about the spatial extension seems speculative.

When modeling the spectra, the proportionality factor entering the inelastic scattering compared to the elastic one seems crucial and should be discussed.

The precise factor entering the calculation is rather meaningless, because the calculation is based on the density of states (DOS) obtained from a lattice model. This accurately describes the on-site DOS, but does not take into account the tunneling matrix element. Hlobil et al. (ref. 26) obtained an inelastic contribution which was larger than the elastic contribution at bias voltages of several gap magnitudes, in agreement with our calculations and experimental findings. It is worth noting that to explain the dip-hump feature by an elastic contribution, a sharp and strong Einstein phonon mode and a strong Van-Hove singularity near E_F are required (see the references for cuprate high- T_c superconductors: PRB 81,214512; PRL 85, 3261; PRL 101,267004), neither of which is found in LiFeAs.

In summary I may not recommend the manuscript in its present form for publication in nature communications due to its somehow speculative conclusion. I also wonder whether the mapping of the spin resonance excitation pinned to an impurity can easily be transferred to that in the bulk material.

LiFeAs is one of the few compounds that has non-polar cleaved surface without surface reconstruction, and where all available evidence from ARPES, STM, and DFT as well as comparison with quantum oscillations show that the surface properties represent the bulk properties quite well. Both, energy as well as the observed length scale, are quite consistent with inelastic neutron scattering, which is a bulk measurement (see discussions in lines 85-89 and lines 149-153 of the

revised manuscript). There is therefore no evidence to suggest that our measurements of spin fluctuations at the surface yield different results compared to the bulk.

The referee does however raise an interesting point, as this might play a role in other materials, where the surface properties might differ from the bulk. This offers a uniquely new perspective, as it will enable comparing how the change in the spin fluctuation spectrum impacts on the superconductivity near the surface. This is however beyond the scope of our study, as in the case of LiFeAs there is no evidence for this kind of effect.

REVIEWERS' COMMENTS:

Reviewer #1 (Remarks to the Author):

The authors have addressed most of the referee's comments, which now make the present manuscript a very nice article. I therefore recommend publication of the revised version in Nat. Comm.

However, there are still a couple of points that I would like the authors to address prior to publication:

1) The authors state on p. 4

"When tunneling into a metal, the lowest available states are right at the Fermi energy and inelastic features will appear once $eV > \Omega$. For a single bosonic mode, this leads to a characteristic step in the conductance that is symmetric in bias due to the opening of an additional inelastic tunneling channel."

I am confused, either the feature appears only when $eV > \Omega$, or it is symmetric in bias, which means that it also appears for $eV < -\Omega$. Both statements cannot be simultaneously true, and the authors should clarify what they mean.

2) The authors should indicate in Eq.(1) that $\chi(\omega)$ is the imaginary part of the spin susceptibility. This is commonly done either by using $\chi'(\omega)$ or $\text{Im } \chi(\omega)$. I know that the authors stated that they used the convention of Hlobil et al. (arXiv:1603.05288), but that convention is not commonly used and therefore can easily lead to confusion.

3) The authors state on p.4:

"Comparison of the simulated tunneling spectrum (Fig. 2d) with those obtained on LiFeAs (see Fig. 3c) show excellent agreement. "

Fig.2d has one curve, Fig.3c has 6 curves. Which curve in Fig.3c shows the excellent agreement with Fig.2d?

Reviewer #2 (Remarks to the Author):

I am satisfied with the authors reply. I recommend publication in Nature Communications

Reviewer #3 (Remarks to the Author):

In their reply and in the revised version the authors strengthened the argumentation that the replica feature can be identified with spin resonance modes mostly studied by neutron scattering experiments, which has been my main criticism and which was also questioned by reviewer #1.

The calculations are very well described in the revised version and the emergence of the feature with the superconductivity can be taken as indication that this feature is related to spin-resonance modes in LiFeAs.

However, I would not consider the agreement between the sharp feature seen in tunneling and the enhancement of magnetic susceptibility in neutron scattering studies as "very good agreement" (page 5). On the contrary the differences between the two techniques should inspire further studies. On the other hand my question about the possibility to transfer these surface results to the bulk properties in LiFeAs was fully answered. I furthermore like the inclusion of the discussion about the more broader relevance of the presented new method suggested by referee #2.

Summarizing I recommend the revised version of the paper for publication in nature communications, because it opens the path to a new possibility to study magnetic excitations in correlated electron systems, that should generate considerable impact.

We thank all reviewers for their constructive reports. We have copied in our replies in *italic* below.

Reviewer #1 (Remarks to the Author):

The authors have addressed most of the referee's comments, which now make the present manuscript a very nice article. I therefore recommend publication of the revised version in Nat. Comm.

However, there are still a couple of points that I would like the authors to address prior to publication:

1) The authors state on p. 4

"When tunneling into a metal, the lowest available states are right at the Fermi energy and inelastic features will appear once $eV > \Omega$. For a single bosonic mode, this leads to a characteristic step in the conductance that is symmetric in bias due to the opening of an additional inelastic tunneling channel."

I am confused, either the feature appears only when $eV > \Omega$, or it is symmetric in bias, which means that it also appears for $eV < -\Omega$. Both statements cannot be simultaneously true, and the authors should clarify what they mean.

The feature appears at both, positive and negative bias, whenever $|eV| > \Omega$. We have clarified this in the revised manuscript.

2) The authors should indicate in Eq.(1) that $\chi(\omega)$ is the imaginary part of the spin susceptibility. This is commonly done either by using $\chi''(\omega)$ or $\text{Im } \chi(\omega)$. I know that the authors stated that they used the convention of Hlobil et al. (arXiv:1603.05288), but that convention is not commonly used and therefore can easily lead to confusion.

Following the suggestion by the referee, we have amended our notation and now explicitly use $\text{Im } \chi(\omega)$.

3) The authors state on p.4:

"Comparison of the simulated tunneling spectrum (Fig. 2d) with those obtained on LiFeAs (see Fig. 3c) show excellent agreement. "

Fig.2d has one curve, Fig.3c has 6 curves. Which curve in Fig.3c shows the excellent agreement with Fig.2d?

This refers to the spectrum taken on the clean surface. For clarity, we now explicitly mention this when referring to figure 3c (now fig. 3h).

Reviewer #2 (Remarks to the Author):

I am satisfied with the authors reply. I recommend publication in Nature Communications

Reviewer #3 (Remarks to the Author):

In their reply and in the revised version the authors strengthened the argumentation that the replica feature can be identified with spin resonance modes mostly studied by neutron scattering experiments, which has been my main criticism and which was also questioned by reviewer #1. The calculations are very well described in the revised version and the emergence of the feature with the superconductivity can be taken as indication that this feature is related to spin-resonance modes in LiFeAs.

However, I would not consider the agreement between the sharp feature seen in tunneling and the enhancement of magnetic susceptibility in neutron scattering studies as "very good agreement" (page 5). On the contrary the differences between the two techniques should inspire further studies. On the other hand my question about the possibility to transfer these surface results to the bulk properties in LiFeAs was fully answered. I furthermore like the inclusion of the discussion about the more broader relevance of the presented new method suggested by referee #2.

Summarizing I recommend the revised version of the paper for publication in nature communications, because it opens the path to a new possibility to study magnetic excitations in correlated electron systems, that should generate considerable impact.

We thank the referee for this assessment, and hope that our work inspires future studies which will hopefully develop a more complete and better description of the relation between what is probed by inelastic tunnelling and what by neutron scattering.